# The Allopregnanolone Response to Acute Stress in Females: Preclinical and Clinical Studies

**DOI:** 10.3390/biom12091262

**Published:** 2022-09-08

**Authors:** Maria Giuseppina Pisu, Luca Concas, Carlotta Siddi, Mariangela Serra, Patrizia Porcu

**Affiliations:** 1Neuroscience Institute, National Research Council of Italy (CNR), 09042 Cagliari, Italy; 2Department of Life and Environment Sciences, Section of Neuroscience and Anthropology, University of Cagliari, 09042 Cagliari, Italy

**Keywords:** allopregnanolone, stress, HPA axis, females

## Abstract

The neuroactive steroid allopregnanolone ((3α,5α)-3-hydroxypregnan-20-one or 3α,5α-THP) plays a key role in the response to stress, by normalizing hypothalamic-pituitary-adrenal (HPA) axis function to restore homeostasis. Most studies have been conducted on male rats, and little is known about the allopregnanolone response to stress in females, despite that women are more susceptible than men to develop emotional and stress-related disorders. Here, we provide an overview of animal and human studies examining the allopregnanolone responses to acute stress in females in the context of stress-related neuropsychiatric diseases and under the different conditions that characterize the female lifespan associated with the reproductive function. The blunted allopregnanolone response to acute stress, often observed in female rats and women, may represent one of the mechanisms that contribute to the increased vulnerability to stress and affective disorders in women under the different hormonal fluctuations that occur throughout their lifespan. These studies highlight the importance of targeting neuroactive steroids as a therapeutic approach for stress-related disorders in women.

## 1. Introduction

It is well accepted that the neuroactive steroid allopregnanolone ((3α,5α)-3-hydroxypregnan-20-one or 3α,5α-THP) plays a key role in the response to stress, by normalizing hypothalamic-pituitary-adrenal (HPA) axis function to restore homeostasis. Most of this evidence comes from research conducted in male rats, and very few studies have focused on allopregnanolone levels in response to acute stress challenges in females, despite the fact that women are more susceptible than men to develop emotional disorders such as depression and anxiety [1,2]. Furthermore, women can suffer from premenstrual syndrome, premenstrual dysphoric disorder (PMDD), postpartum depression, or perimenopausal depression, all conditions related to the hormonal fluctuations associated to their reproductive function.

The aim of this review is to provide an overview of animal and human studies examining the allopregnanolone responses to acute stress in females and to highlight their relevance in the context of stress-related neuropsychiatric diseases.

## 2. HPA Axis, Role of GABA, Actions of Allopregnanolone

The development and termination of a challenge stress response are equally critical aspects of homeostatic and allostatic regulation. The endocrine stress response involves direct actions of corticotropin releasing hormone (CRH) and arginine vasopressin, produced by hypothalamic neurons in the paraventricular nucleus (PVN), to anterior pituitary corticotropes to drive adrenocorticotropic hormone (ACTH) release into systemic circulation. The increase in circulating ACTH stimulates the synthesis and secretion of glucocorticoids (cortisol in humans and corticosterone in rodents) from the adrenal cortex, which, in turn, activate glucocorticoid receptors widely expressed in the central nervous system [3]. Activation of the HPA axis is limited by glucocorticoid negative feedback inhibition in the PVN acting through an endocannabinoid-dependent mechanism to rapidly reduce both neural activity and glucocorticoid release [4,5]. The involvement of endocannabinoids in the control of the glucocorticoid stress response is demonstrated by the evidence that the dexamethasone fast feedback response is blocked by the cannabinoid CB1 receptor antagonist AM-251 [6]. The γ-aminobutyric acid (GABA) neurotransmission has also an important role in HPA axis inhibition: more than one third of inputs to CRH neurons in the medial parvocellular PVN are from GABAergic neurons located in the immediate surround [7] and involve cortical and limbic forebrain regions [8]. Prefrontal and hippocampal neurons play a major role in HPA axis inhibition mediated by hypothalamic and bed nucleus of the stria terminalis GABAergic relays to the PVN [9]. The anatomical studies are consistent with multiple pharmacological evidence that reveals a tonic inhibition of the HPA axis and the capability of this system to control the stress response [9]. By virtue of its role as a potent endogenous modulator of type A receptors for GABA (GABA_A_ receptors), through which it exerts several psychopharmacologic actions, such as anxiolytic, antidepressant, anticonvulsant, sedative, analgesic, and amnesic effects [10], the neuroactive steroid allopregnanolone provides local inhibition and a long-loop negative feedback to the HPA axis, therefore playing a crucial role in the response to stress. Acute stress induces a marked increase in allopregnanolone concentrations in brain and plasma of male rats [11,12,13], as well as in human serum [14]. Such elevations in allopregnanolone levels exert negative feedback upon the HPA axis, thus reducing the neuroendocrine response to stress by inhibiting CRH production and release, ACTH release, and the subsequent corticosterone increase in rats subjected to acute stress [15,16]. In addition, allopregnanolone may regulate the response to stress by a direct action in the brain; in fact, acute stress increases allopregnanolone levels in cerebral cortex, but not plasma, of adrenalectomized rats, suggesting that the cerebral cortex preserves its neurosteroidogenic properties in response to stress, independent from activation of the HPA axis [13].

We and others have hypothesized that the increase in levels of allopregnanolone in the brain represents a homeostatic mechanism necessary to restore the GABAergic tone altered by acute stress [17,18]. This hypothesis is based on studies mostly conducted in male rats. However, divergent neuroactive steroid responses to acute stress have been found depending on type of stressor, strain of rodents and species [19]. Are there also sex differences? The following sections will address the divergent allopregnanolone responses to stress in females.

### Effect of Acute Stress on Allopregnanolone Levels in Female Animals

Different effects of acute stress on allopregnanolone levels in females have been reported depending on type of stressor, species, other environmental conditions, or brain vs. peripheral levels. We started to pay attention to this topic when we failed to observe the “expected” allopregnanolone increase in response to acute foot shock stress in control female Sprague-Dawley rats under different experimental paradigms. In fact, control females subjected to foot shock stress failed to show an increase in brain and plasma allopregnanolone levels 30 min after stress [20,21], as it was expected from previous results in male rats [12], and despite the normal stress-induced corticosterone response. By contrast, under conditions of reduced basal allopregnanolone levels, such as those induced by a single exposure to oestradiol benzoate on the day of birth, female rats showed the “expected” allopregnanolone increase in response to acute foot shock stress [21]. We hypothesized that the elevated allopregnanolone levels in control females compared to males may provide protection against the effects of acute stress. In agreement with this hypothesis, administration of progesterone to male Sprague-Dawley rats increased allopregnanolone content to levels like those in females and abolished its stress-induced increased concentrations [21]. However, the effect of acute stress on allopregnanolone content may also depend on the nature of the stressor used. In fact, a psychological stressor such as restraint can increase allopregnanolone levels in females of the same strain, while a physical stressor (such as foot shock) does not (Table 1) [22]. In agreement with this interpretation, swim stress was found to increase allopregnanolone content in the frontal cortex, amygdala, and brainstem of female Sprague-Dawley rats [23], although this result was not replicated in a later study where swim stress did not increase plasma or brain allopregnanolone content in control females [24]. Furthermore, stimulation of the HPA axis by CRH or ACTH challenges increased plasma, but not brain, allopregnanolone levels in female Wistar rats, while the same challenges markedly elevated allopregnanolone content in males [25]. Moreover, increased plasma allopregnanolone levels were observed in female C57BL/6J mice subjected to different physical (tail suspension) or psychological stressors (restraint and predator odour stress) [26]; however, these mice had also been exposed to ethanol consumption, a variable known to alter allopregnanolone levels [19,27].

It is important to point out that when allopregnanolone is exogenously administered into the brain, it can restore the HPA axis response to acute stress in sheep [28]. Although endogenous allopregnanolone levels were not measured in these animals, these results further emphasize the therapeutic utility of this neuroactive steroid against stress-related disorders [29,30,31].

## 3. HPA Axis Dysregulation in Neuropsychiatric Diseases

A dysregulation of the HPA axis occurs following chronic stress in both rodents and humans, resulting in blunted elevations in corticosterone/cortisol and allopregnanolone levels. Similar effects are also observed in psychiatric illnesses, including depression, anxiety, post-traumatic stress disorder, or alcohol use disorders [27,32,33,34]. Alterations in CRH, ACTH and cortisol levels have been reported in depressed subjects [32,35], along with reduced sensitivity to glucocorticoids [36], and an imbalance in mineralocorticoid (MR) and glucocorticoid (GR) receptors [37,38]. Likewise, decreased allopregnanolone levels were reported in plasma or cerebrospinal fluid from patients with major depression, anxiety, PMDD, post-traumatic stress disorder, or alcohol use disorders [39,40,41,42,43,44,45], and restoration of neuroactive steroid levels has been proposed as a useful therapeutic approach [27,29,30,46,47,48], suggesting that neuroactive steroids may contribute to the aetiology of affective disorders.

### Effect of Acute Stress on Allopregnanolone Levels in Females under Chronic Stress

In rodents, brain and plasma allopregnanolone levels are reported to be 2–16-fold higher in females than males [20,23,49,50], likely due to the difference in the precursor steroid supplied via blood circulation between females and males [23]. Our group demonstrated a marked reduction in the central and peripheral synthesis of allopregnanolone in socially isolated female rats [20]; although to a less extent, these changes are similar to those observed in male rats [51]. The main symptoms caused by social isolation stress in rodent studies have been correlated with symptoms of neuropsychological disorders, in particular anxiety, depression, schizophrenia, and post-traumatic stress disorder [52,53,54,55], although clinical studies have shown lower, similar, or even higher concentrations of allopregnanolone in women suffering from these and other affective disorders. It is a common opinion that women are more vulnerable to stress; in particular, women suffering from mood disorders show a greater daily stress and a greater rate of traumatic stress [2,56]. Considering that the allopregnanolone increase in response to acute stress represents a homeostatic mechanism in the context of adaptation to stress by limiting the extent and duration of reduction in the GABAergic inhibitory transmission and activation of the HPA axis [17], it is important to understand if there is a correlation between the stress response, the allopregnanolone response and the manifested symptoms. In male rats, the increase in plasma and brain allopregnanolone levels appears after both the stressor onset and the reduction in GABAergic neurotransmission, resulting in a functional correlation among brain allopregnanolone concentrations, recovery of the GABAergic transmission, and recovery from the anxiety-like behaviour [57]. In the social isolation model, despite the expected increase in corticosterone levels, females did not show the increase in plasma and brain levels of allopregnanolone after foot shock stress [20]; however, as mentioned above, emotional stressors such as forced swim [23] or restraint stress [22] significantly increased plasma allopregnanolone levels and such increase was more pronounced in socially isolated female rats (Table 1). It has been suggested that an insufficient allopregnanolone response to stress may predispose individuals to the negative effects of stress [58], perhaps creating the basis for the development of mood disorders [59]. Women with a history of depression suffering from PMDD showed a blunted allopregnanolone response to venipuncture or to the psychosocial Trier Social Stress Test [60]. Women with post-traumatic stress disorder, who show a marked reduction in cerebrospinal fluid levels of allopregnanolone + pregnanolone during the follicular phase [41], failed to significantly increase the ratio of allopregnanolone + pregnanolone to 5α-dihydroprogesterone in response to a moderately stressful fear-conditioning laboratory task [61], suggesting they are uncapable to show the allopregnanolone increase in response to stress.

## 4. Allopregnanolone Levels Following Acute Stress in Women

Studies in women report a plethora of different results, depending on several variables and histories of disease. Here, we provide an overview of allopregnanolone responses to acute stress under different conditions that characterize the female lifespan associated with the reproductive function (Figure 1). Women have a higher prevalence of stress-related disorders, including major depressive disorder and anxiety, and hormonal fluctuations associated with reproductive function may account for this increased vulnerability to stress-related disorders, beginning with a heightened risk of developing a depressive episode during adolescence [62]. Indeed, an increased prevalence of these disorders is observed during hormonal variations occurring at puberty, the premenstrual period, pregnancy, postpartum and perimenopause [1,63,64]. In addition, the hormonal milieu in women is further influenced by administration of exogenous steroids such as those in the hormonal contraceptive formulations [22,65]. All these conditions alter basal circulating allopregnanolone levels and differentially affect its response to acute stress.

### 4.1. Puberty

Puberty marks the beginning of reproductive function with the appearance of the ovarian cycle and the secretion of oestradiol and progesterone from the ovaries. This developmental stage also marks a change in HPA axis function and stress reactivity as the enhanced and prolonged corticosterone response to stress observed in prepubertal female rats normalizes towards adult levels [66]. Basal allopregnanolone levels also increase throughout puberty both in the cerebral cortex of female rats [67] and in girls’ serum [68], although a decline in hippocampal levels has been reported in female C57BL/6 mice at the onset of puberty [69]; however, to the best of our knowledge, allopregnanolone levels in response to acute stress have not yet been assessed in pubertal females. Nonetheless, the abrupt reduction in hippocampal allopregnanolone at the onset of puberty has been related to a paradoxical increased anxiety in pubertal female C57BL/6 mice, likely driven by a decrease in the inhibitory tonic current mediated by extrasynaptic α4βδ GABA_A_ receptors [69]. Overall, this result suggests that the abrupt changes in allopregnanolone levels that occur at puberty may contribute to the increased emotional disorders in female adolescents; however, whether and how acute stress alters allopregnanolone levels in adolescent females remains to be investigated.

### 4.2. Menstrual Cycle

Allopregnanolone fluctuates across the menstrual cycle in women, with higher levels observed during the luteal phase [45,70]. Likewise, higher allopregnanolone levels were observed during the metoestrus/dioestrus 1 phase of the rodent oestrous cycle [71,72]. Fluctuations in allopregnanolone content throughout the menstrual cycle have been linked to mood symptoms during the premenstrual period and to the development of premenstrual syndrome and PMDD [29,73]. Similarly, sensitivity to stress, assessed via cortisol secretion or cardiovascular reactivity, increases in the luteal phase of the menstrual cycle [74,75], and may also contribute to these mood symptoms. Does allopregnanolone play a role in this response? Brain imaging studies showed that a greater allopregnanolone increase during the luteal phase was associated with smaller amygdala and prefrontal cortical responses to a mild psychological stress, suggesting that fluctuations in allopregnanolone levels across the menstrual cycle may influence vulnerability to stress [76]. However, this study only measured basal allopregnanolone in the follicular vs. the luteal phase, but not after stress exposure.

Pharmacological activation of the HPA axis with a CRH challenge increased allopregnanolone content in healthy controls during the follicular phase [77], while allopregnanolone levels were increased following an ACTH challenge and decreased following a dexamethasone challenge in healthy controls tested during the luteal phase [78]. Likewise, the psychosocial Trier Social Stress Test increased allopregnanolone levels in women during the luteal phase of the menstrual cycle, but the same condition failed to alter allopregnanolone levels during the follicular phase, despite the expected increase in cortisol levels [79,80]. Thus, the increase in allopregnanolone content in response to stress is not always observed in women. These results are limited by the small number of subjects involved in the studies, but it appears that several variables may influence the allopregnanolone response to stress. One such variable can be ethnicity; in fact, in women tested during the luteal phase, the psychosocial Trier Social Stress Test did not increase allopregnanolone levels overall, but this effect was related to ethnicity; 59% of non-Hispanic white women showed the stress-induced increase in allopregnanolone levels, but among African American women this percentage dropped to 20%, and acute stress decreased allopregnanolone levels in this latest group [81]. Overall, the allopregnanolone and cortisol responses to this acute stress were inversely correlated in women; however, these effects were not observed in men, as no ethnic differences in the allopregnanolone response to stress were present [81].

Nonetheless, several studies point to a contribution of the allopregnanolone response to stress in the development of premenstrual syndrome and PMDD. For example, Girdler and collaborators examined the effects of acute psychosocial stress (speech stress and paced auditory serial addition test) in patients with PMDD during the luteal phase of the menstrual cycle [82]. These patients showed greater baseline allopregnanolone levels compared to healthy controls, but had a blunted allopregnanolone response to stress, while healthy controls showed a mild stress-induced increase in allopregnanolone content. Of note, cortisol levels increased in both controls and PMDD women following acute stress. Furthermore, PMDD women with greater levels of premenstrual anxiety also showed lower basal allopregnanolone levels compared to less anxious patients [82]. This result agrees with subsequent evidence for a reduced basal allopregnanolone content in women suffering from premenstrual syndrome, and a blunted response to both the ACTH and dexamethasone challenges, compared to healthy controls, all tested during the luteal phase [78].

A similar finding was reported when examining the effects of prior history of depression on the allopregnanolone response to stress in PMDD [60]. Women with a prior history of depression, regardless of PMDD symptoms, showed a blunted allopregnanolone response to stress (both at 30- and 60-min post stress), while the mild increase in circulating allopregnanolone levels in those women with no history of depression was not significant. In addition, women with a history of depression showed a prolonged allopregnanolone response to stress, compared to those with no history of depression, suggesting that history of depression (even in the absence of current depression) causes a failure in the allopregnanolone ability to respond to challenges. Furthermore, a greater blunted allopregnanolone reactivity to mental stress predicted worse PMDD symptoms, but only in those women with prior depression [60].

### 4.3. Pregnancy

During pregnancy, brain and peripheral allopregnanolone levels rise dramatically, reaching a peak in late pregnancy in both rats and women [83,84], and these sustained levels might contribute to the blunted HPA response to stress. In fact, the HPA axis in pregnant females is less reactive, resulting in blunted CRH, ACTH and corticosterone responses to different psychological and physical stressors, an adaptation apt to protect the foetus [85,86]. The allopregnanolone response to stress also appears to be reduced in women subjected to a psychosocial stress on the second trimester of pregnancy, as such stress resulted in blunted cortisol, progesterone and allopregnanolone + pregnanolone levels; furthermore, lower allopregnanolone and allopregnanolone + pregnanolone responses were associated with greater negative emotional responses to the stress tests [87].

Several factors may contribute to the mechanisms that lead to HPA axis hypo-responsiveness during pregnancy [88], and allopregnanolone may play a critical role in these adaptations. One example of the allopregnanolone involvement is the induction of the inhibitory opioid mechanism in late pregnancy that prevents activation of CRH neurons, and the consequent suppressed response to an immune challenge [89]. In late pregnancy, the HPA axis responses to stressors, including an immune challenge with interleukin-1β, are attenuated by a central opioid mechanism that prevents noradrenaline release in the PVN; this effect can be reversed by inhibition of 5α-reductase, suggesting that allopregnanolone is mediating such mechanism [90]. Indeed, allopregnanolone administration reduced the HPA responses to the immune challenge in virgin rats [90], in agreement with its involvement on restoration of HPA homeostasis under basal healthy conditions.

In addition to this mechanism, the sustained elevation in allopregnanolone levels throughout pregnancy alters GABA_A_ receptor subunit expression and function [83,91,92], and the resulting changes in GABAergic inhibition may likely contribute to the blunted HPA response to acute stress.

### 4.4. Postpartum

The sustained elevation in brain and peripheral allopregnanolone levels during pregnancy abruptly declines immediately before parturition and its levels remain low during the postpartum period [83]. Such fluctuations may account for development of postpartum depression in vulnerable women [93] and indeed brexanolone, the proprietary formulation of allopregnanolone, has been the first neuroactive steroid to have gained approval for treatment of this disorder [94]. Stress also contributes to postpartum depression, but the allopregnanolone contribution to the HPA response to stress during postpartum is poorly explored. The psychosocial Trier Social Stress Test activated the HPA axis during postpartum, resulting in increased ACTH and cortisol responses 10 min after the completion of the test in lactating and non-breast feeding women at postpartum, as well as in controls during the follicular phase; however, the same stress failed to alter circulating allopregnanolone levels in all these groups tested [80], indicating once again that the allopregnanolone response to acute stress does not always parallel the HPA activation. The effects of a single acute stress on allopregnanolone content in postpartum rodents have not been explored. By contrast, pup separation in postpartum days 3 to 15 induced an increase in plasma allopregnanolone levels in female Sprague-Dawley rats tested at postpartum day 21, compared to naïve dams; this increase was abolished by exposure to daily intubation during pregnancy, likely a consequence of the HPA axis adaptations to this chronic stress [95]. Thus, the allopregnanolone response to stress during postpartum is not strictly related to its fluctuations associated with parturition and lactation, but it may depend on the overall stress experience of the individual and subsequent HPA adaptions.

### 4.5. Perimenopause

Perimenopause is a stage of transition from the reproductive age to senescence, lasting 1–5 years and characterized by fluctuating levels of oestradiol and progesterone as the ovaries gradually cease their function; such hormonal fluctuations can contribute to changes in stress responsiveness and increased risk of depression, one the several symptoms associated with perimenopause [96]. Basal allopregnanolone levels either do not change with aging (compared to those during the follicular phase) [70] or decrease in menopausal women [97]; however, its responses to acute stress have not yet been examined at this stage. Nonetheless, it has been hypothesized that fluctuations in allopregnanolone levels during perimenopause might contribute to insufficient plasticity of GABA_A_ receptors and to their ability to control HPA axis function, thus leading to increased risk for depression in vulnerable women [63]. Future studies are required to further explore this hypothesis.

## 5. Hormonal Contraceptives and Allopregnanolone

Hormonal contraceptives offer effective and reversible regulation of fertility, representing one of the most reliable methods for birth control used by 43% of women of reproductive age worldwide [98]. These drugs also have numerous non-contraceptive benefits related to women’s health [99]. Nevertheless, use of hormonal contraceptives may exert some psychological effects; indeed, occurrences of depression or mood changes are among common reasons for stopping effective hormonal contraceptive use [100,101]. However, effects of hormonal contraceptives on mood disturbances are still a matter of debate, as both mood worsening and mood improvement or stabilization have been reported [65,102,103,104,105,106,107,108,109,110,111]. Among the risk factors that predispose hormonal contraceptive users to negative mood changes are personal and family histories of depression, postpartum depression, premenstrual syndrome, and young age [107,112,113,114,115].

The active components of hormonal contraceptives are synthetic oestrogen and progestin compounds, usually available as oestro-progestin formulations. They prevent ovulation by inhibiting the hypothalamic-pituitary-ovarian axis, thus reducing circulating levels of endogenous oestradiol, progesterone and allopregnanolone that occur during the menstrual cycle [99,107,116,117]. In agreement, chronic treatment with ethinyl oestradiol (EE) and levonorgestrel (LNG), two of the synthetic steroids most widely used in the oestro-progestin combinations of hormonal contraceptives, markedly reduced brain and plasma concentrations of allopregnanolone in female rats [116,118,119,120]. Such decrease is associated to an anxiety-like behaviour in the elevated plus-maze test [116,118], and to a blunted response to stress [22]. In fact, acute restraint stress increased plasma allopregnanolone levels in vehicle-treated female rats but did not significantly change such levels in EE-LNG-treated female rats [22], likely due to alterations in the HPA axis function. In agreement, chronic treatment with EE-LNG increased basal plasma corticosterone levels in female rats [22], an effect that might be related to the adaptations in HPA axis function elicited by reduced basal allopregnanolone levels in these rats. Furthermore, chronic EE-LNG treatment increased the expression of hippocampal MR, but not GR, in the same animals (Porcu et al., unpublished observation). GR and MR mainly mediate the threshold setting and the extinction of the stress response, and an imbalance between these receptors alters HPA axis function and sensitivity to stress [37,38]. Accordingly, acute exposure of female rats to restraint stress induced a marked increase in plasma corticosterone levels both in vehicle-treated and EE-LNG-treated rats; however, the increase in corticosterone levels induced by restraint stress in EE-LNG-treated rats was significantly lower than that observed in vehicle-treated rats, suggesting that hormonal contraceptives attenuated the HPA response to acute stress [22]. This result agrees with the observations reported in women; in fact, use of hormonal contraceptives increases basal cortisol concentrations in women [117,121,122], and alters the HPA axis response to stress, resulting in a blunted increase in cortisol levels in response to both psychosocial stress and pharmacological stimulation of the HPA axis with naltrexone and ACTH [121,122,123,124,125]. Moreover, an attenuation in plasma ACTH elevations following stimulation with CRH has also been reported in hormonal contraceptives users [126].

The blunted allopregnanolone and corticosterone responses to restraint stress following chronic EE-LNG treatment may increase vulnerability to develop alterations in the emotional state in rats and hence mood disorders in predisposed hormonal contraceptives users. However, the allopregnanolone response to stress in women using hormonal contraceptives has not yet been evaluated. Hypothalamic amenorrhea induced a condition similar to that of hormonal contraceptives, as women with this disease showed reduced basal allopregnanolone, increased basal cortisol levels, and a blunted allopregnanolone response to CRH challenge [77]. However, the involvement of this neuroactive steroid in the regulation of the HPA response to acute stress following hormonal contraceptives exposure remains to be clarified. We would have predicted that the reduced basal levels of allopregnanolone following hormonal contraceptives exposure might have resulted in an hyperstimulation of the HPA response to stress to restore homeostasis, similar to the response observed under conditions of chronic stress (i.e., the social isolation model in female rats [20]); instead, we observed the opposite outcome. Regulation of HPA homeostasis is under strict hormonal control and the marked hormonal changes induced by hormonal contraceptives may affect the HPA axis in ways that are only partially dependent from allopregnanolone regulation. For instance, hormonal contraceptives, by blocking ovulation, also markedly reduce the content of oestradiol, which as well plays a role in the response to stress. Oestrogen administration increases basal corticosterone secretion as well as ACTH and corticosterone responses to physical and psychological stressors [127]. Likewise, in ovariectomized rats, a condition in which the hormonal milieu is similar to that induced by hormonal contraceptives, activation of the HPA axis in response to stress is blunted [128,129]. Thus, the allopregnanolone regulation of HPA homeostasis may be influenced by the global hormonal milieu in the whole organism. Furthermore, women carriers of the MR-haplotype 1 (MR-2 G and MRI180V A polymorphisms, associated with an altered MR expression [130]) are more sensitive to the depressogenic effects of hormonal contraceptives [131,132], suggesting that changes in MR expression might also underlie the reduced sensitivity to stress caused by hormonal contraceptives. In summary, future studies are required to uncover the neurobiological mechanisms involved in the blunted HPA response to acute stress induced by hormonal contraceptives and the putative role for allopregnanolone in such response.

## 6. Concluding Remarks

Appropriate responses to stress promote survival by altering physiological processes and behaviour, and the potential role for allopregnanolone in the fine-tuning and cessation of the stress response has been supported by studies mainly conducted in male rats [13,15,16,17]. We reviewed evidence showing that this mechanism does not always generalize to females. Under certain circumstances, for instance when basal allopregnanolone levels are elevated, or under conditions of abrupt fluctuations, females have a blunted allopregnanolone response to acute stress. In agreement, elevated allopregnanolone following its systemic administration did not affect CRH expression in the hypothalamus and hippocampus of female rats [133,134]. Likewise, allopregnanolone has been shown to have a paradoxical effect in PMDD that can be reversed by GABA_A_ receptor modulating steroid antagonists [29]. This evidence suggests that allopregnanolone supplementation per se might not always be an optimal therapeutic approach. However, the hypothesis that the blunted allopregnanolone response to acute stress may represent one of the mechanisms that contribute to the increased vulnerability to stress and affective disorders in women under the different hormonal fluctuations that occur throughout their lifespan, is still valid. Therefore, allopregnanolone or its analogues may have therapeutic utility in all those conditions where this neuroactive steroid fails to regulate the response to stress [30,31].

## Figures and Tables

**Figure 1 biomolecules-12-01262-f001:**
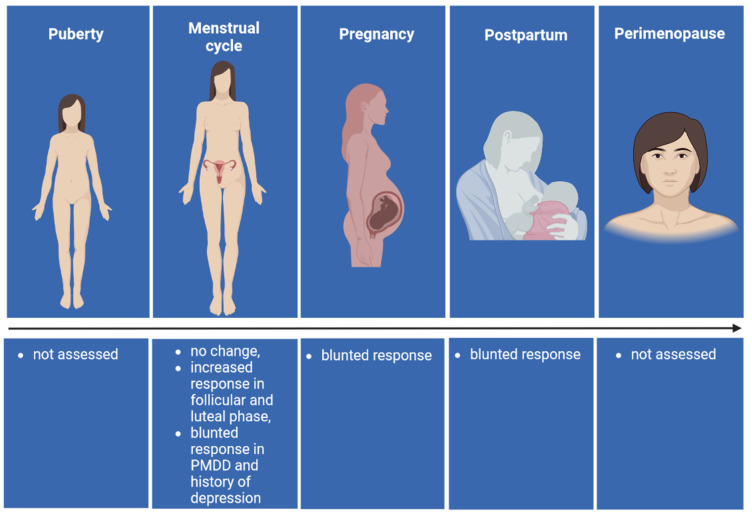
Schematic representation of the allopregnanolone response to acute stress under different conditions that characterize the female lifespan. Figure created using BioRender.com accessed on 27 July 2022.

**Table 1 biomolecules-12-01262-t001:** Effect of physical and psychological acute stressors on plasma allopregnanolone and corticosterone levels in group-housed and socially isolated female rats.

Experimental Group	Foot Shock Stress	Restraint Stress
**Allopregnanolone**		
Group-housed	14.3 ± 1.2	15.7 ± 2.0
Group-housed + acute stress	14.9 ± 1.8	25.8 ± 3.4 ^a^
Socially isolated	9.8 ± 0.8 ^a^	10.2 ± 1.3 ^b^
Socially isolated + acute stress	10.0 ± 1.1	25.7± 3.6 ^c^
**Corticosterone**		
Group-housed	132.1 ± 12.0	151.2 ± 9.9
Group-housed + acute stress	190.2 ± 17.3 ^b^	1343.9 ± 140.3 ^a^
Socially isolated	70.1 ± 8.8 ^b^	72.0 ± 11.2 ^b^
Socially isolated + acute stress	324.6 ± 42.0 ^de^	1770.1 ± 261.1 ^de^

Socially isolated or group-housed female rats were exposed to acute stress procedures (foot shock stress, 5 min; restraint stress, 10 min), and were sacrificed 30 min after beginning of the stress exposure for measurement of plasma allopregnanolone and corticosterone. Data are expressed as ng/mL of plasma and are means ± SEM of values from 15 rats per group. ^a^
*p* < 0.05, ^b^
*p* < 0.01 vs. non-stressed group-housed animals; ^c^
*p* < 0.05, ^d^
*p* < 0.01 vs. non-stressed socially isolated animals; ^e^
*p* < 0.01 vs. the respective stressed group-housed animals (two-way ANOVA followed by Newman–Keuls post hoc test).

## Data Availability

Not applicable.

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
