# Peer review of "The Allopregnanolone Response to Acute Stress in Females: Preclinical and Clinical Studies"

_biomolecules, 2022, doi:10.3390/biom12091262_

Round 1

Reviewer 1 Report

This review entitled “The allopregnanolone response to acute stress in females: Preclinical and clinical studies” reports interesting review on steroid drug allopregnanolone as an antidepressant drug application in female. The manuscript explored an overview of animal and human studies examining the allopregnanolone responses to acute stress in females in the context of stress-related neuropsychiatric diseases and under the different conditions that characterize the female lifespan associated with the reproductive function.

1.     Table 1. What could be reason that socially isolated and socially isolated + acute stress shows lower allopregnanolone concentration after foot shock stress?

2.     Line 137-141, Decreased level of allopregnanolone in plasma of patients with major depression and stress is contradicting results than rats/mice. Why?

3.     Line 244-249, What is the conclusion of such huge variation in concentration of allopregnanolone? How is the variation in concentration of allopregnanolone reliable for stress diagnosis?

4.     Conclusion is remarkable.

Reviewer 2 Report

It is an excellent overview of animal and human studies examining the allopregnanolone responses to acute stress 14 in females in the context of stress-related neuropsychiatric diseases and under the different conditions.  I only have little concerns:

1. Some of the references are too old, can you replace them with literature published in  the last five years?

2. Figure 1 is not clear, can you provide a better one?

3. Can add a figure to summarise your findings.
